# Diversity and Composition of Rumen Bacteria, Fungi, and Protozoa in Goats and Sheep Living in the Same High-Altitude Pasture

**DOI:** 10.3390/ani10020186

**Published:** 2020-01-22

**Authors:** Suo Langda, Chenguang Zhang, Ke Zhang, Ba Gui, De Ji, Ciren Deji, Awang Cuoji, Xiaolong Wang, Yujiang Wu

**Affiliations:** 1Institute of Animal Sciences, Tibet Academy of Agricultural and Animal Husbandry Sciences, Lhasa 850009, China; Sonaada10@163.com (S.L.); basang125@163.com (B.G.); deji2019@163.com (D.J.); crdj464036137@163.com (C.D.); 18392377957@163.com (A.C.); 2College of Animal Science and Technology, Northwest A&F University, Yangling 712100, China; chenguangzhang@aliyun.com (C.Z.); kezhang@nwafu.edu.cn (K.Z.)

**Keywords:** amplicon gene sequencing, adaptive evolution, high altitude, rumen microbes, ruminant, volatile fatty acids

## Abstract

**Simple Summary:**

Tibetan goats and sheep graze together but have different growth performances, immune responses, and feeding preferences in the Tibetan pasture. Rumen microbiota composed of bacteria, fungi, and protozoa are necessary for a healthy ruminant. Therefore, in this study, we comprehensively describe composition and diversity of bacteria, fungi, and protozoa in the high- altitude rumen. Compared with sheep, the bacteria that degrade crude protein and produce volatile fatty acids (VFA) were increased in the rumen of goats (Saccharofermentans and Lachnospiraceae_XPB1014) (*p* < 0.05). In addition, when compared with goats, the fungi and protozoa that degrade fiber were increased in rumen of sheep (Neocallimastigaceae and Metadinium) (*p* < 0.05). Furthermore, VFA were significantly increased in the rumen of goats compared with sheep (*p* < 0.05). The VFA level was consistent with differences in the microbiota composition in the rumen between goats and sheep. Under mixed grazing conditions, goats tend to select a high-crude protein diet that is good for growth, whereas sheep tend to select a high-lignin diet that is difficult to digest. Therefore, the different microbiota in the rumen of goats and sheep may be explained by dietary preference.

**Abstract:**

Environmental adaptation of ruminants was highly related to microbiota in the rumen. To investigate the diversity and composition of bacteria, fungi, and protozoa in the rumen of high-altitude animals, amplicon gene sequencing was performed using rumen fluid samples derived from both Tibetan goats and sheep at the same pasture in a highland (altitude > 4800 m). Between these two species, the ruminal bacteria and fungi were significantly different at multiple taxonomic levels. The alpha diversity of bacteria was significantly high in goats (*p* < 0.05). One hundred and sixty-four and 29 Operational Taxonomy Units (OTUs) with significant differences were detected in bacteria and fungi, respectively. The abundance of bacteria, fungi, and protozoa in the rumen was characterized at multiple taxonomic levels, and we determined that Firmicutes, Bacteroidetes, Neocallimastigomycota, and Ciliophora were the most abundant bacteria, fungi, and protozoa. The family Neocallimastigaceae and the genus Metadinium had cellulose degradation capacity in the rumen with high abundance, thereby, suggesting that fungi and protozoa played an essential role in rumen fermentation. In addition, by comparing microbiota in the rumen of goats and sheep it was found, that the fiber-degrading fungi genus (Cyllamyces) was increased in the rumen of sheep (*p* < 0.05) whereas VFA-producing bacteria (Saccharofermentans and Lachnospiraceae_XPB1014) were increased in the rumen of goats (*p* < 0.05). Interestingly, in the rumen, no differences in protozoa were observed between goats and sheep (*p* > 0.05). Furthermore, when compared to sheep, level of acetic acid, propionic acid, and total volatile fatty acid (TVFA) were significantly increased in the rumen of goats (*p* < 0.05). Taken together, these results suggested microbiota in the rumen drive goats to better adapt to high-altitude grazing conditions.

## 1. Introduction

Tibetan goats and sheep are important livestock in the Tibetan Plateau of China, which provide sustenance for pastoralists and products for trade. Ruminants, such as goats and sheep were domesticated about 11,000 years ago, accompanied by human migration, especially by the nomads spreading to the world [1]. During domestication and proliferation, goats, and sheep were domesticated to various degrees, and their unique environmental adaptability was formed [2]. In a previous study, small-tailed Han sheep were introduced to the plateau and their digestive and metabolism energy between Tibetan sheep and small-tailed Han sheep was compared. The data showed that as the gross energy of the diet decreased, Tibetan sheep showed the well-adapted capability for plateau, when compared to small-tailed sheep [3]. Regarding living habits, due to the dietary conditions, and adaptive anatomical and physiological mechanisms [4], goats were fed a wide variety of foods, including multifarious pastures, crop straws, shrubs, young leaves, roots, tubers, agricultural and sideline products, food-processed skins, shells and slags [5]. In a previous study on the feeding preferences of Tibetan goats it was shown that in summer (grassy period), Tibetan goats preferred Carex mpprcroftii and Kobresia pygmaea, whereas in the fall, the intake of Polentilla bifurca and Anaphalis xylorrhiza was increased [6]. Although goats are known for their ability to digest browsers or fibrous materials, they tend to regulate browsers or fibrous materials intake in environmental pressure, whereas free range sheep would continuously feed on fibrous materials [7]. Compared to sheep, the thickness of small intestinal and muscularis mucosa was higher in goats [5], thereby suggesting that feeding preference and physiological structure existed difference between goats and sheep.

Rumen microbial community can be shaped by dietary, adaptive anatomical, and physiological mechanisms and has evolved together with the variety of feeding strategies in various ruminant lineages [8]. In a previous study, it was found that compared to sheep, goats had a higher rumen ammonia level [9]. Notably, in a certain range, higher ruminal ammonia-N concentration was more optimal for efficiency of microbial growth and led to higher VFA concentration [3]. Metagenomic sequencing showed that microbiota genes in the rumen in high-altitude ruminant were significantly enriched in VFA-yielding pathways, and that the genes at low-altitudes were significantly enriched in methanogenesis pathways. These data suggested that when compared to low-altitude ruminants, high-altitude ruminants have lower methane emissions and higher feed conversion ratio [10]. Rumen microbiota could also increase the plant variety for ruminant feeding. The infusion of a tannin solution into the rumen significantly decreased the level of intake of sheep but had no effect on goats [11]. The ability of goats to tolerate tannins was not only related to the high level of parotid salivary secretion [12] and the amount of recycled urea [13], but was also linked to the presence of tannin-degrading bacteria (Selenomonas ruminantium) in the rumen of goats [9,14]. Together, the above-mentioned studies suggested that rumen microbiota might play an important role in the metabolic capacity of the rumen of goats and sheep.

The Tibetan Plateau is featured with a distinct grazing ecosystem. Tibetan goats and sheep, an important income source for generations, inhabit and proliferate in the special ecological area with extensive management. Therefore, the area is an ideal place to examine the microbe diversity and composition in sheep and goats that live in the high-altitude pasture. The aims of this study were to characterize high-altitude rumen microbiota and to identify the influencing factors that affect the quality of the rumen microbiota between different hosts.

## 2. Materials and Methods

### 2.1. Animal Handling, Diet, and Sampling

Grazing goats and sheep were raised in the Qiangtang Plateau (Nagchu, Tibet, China; altitude more than 4800 m). A total of forty goats and sheep (n = 20 each) were subjected to simultaneous estrus and artificial insemination. After the goat kids and lambs were born, they followed the dam for grazing together, and no artificial feeding was used. One-year-old female kids and lambs (n = 5 each) were randomly selected to collect rumen fluid samples using oral stomach probing. To reduce invasion of the body, hot water-heated Poly Vinyl Chloride pipes were inserted to a depth of approximately 120–150 cm through the esophagus at 3 h after feeding, then the rumen contents were immediately collected using a vacuum pump, to avoid possible contamination of the samples with saliva. The samples (approximately 50 mL) were strained by a nylon membrane, then, samples were immediately frozen in liquid nitrogen for further study [15].

All experimental procedures were approved by the Institutional Animal Care and Use Committee of the Northwest A&F University under permit number 2014ZX08008002.

### 2.2. DNA Extraction, PCR Amplification, and Sequencing

Microbial DNA was extracted by the E.Z.N.A.^®^ DNA Kit (Omega Bio-tek, Norcross, GA, USA) from 10 ruminal content samples. The bacterial region (V3–V4) of the 16S rRNA gene was amplified by 338F (5′-CCTAYGGGRBGCASCAG3′) and 806R (5′-GGACTACHVGGGTWTCTAAT-3′) primers [16]. Moreover, the fungal ITS rRNA gene was amplified using ITS1F (5′-CTTGGTCATTTAGAGGAAGTAA-3′) and ITS2R (5′-GCTGCGTTCTTCATCGATGC-3′) primers [17]. The protozoa were identified by 18S rRNA gene sequencing using RP841F (5′-GACTAGGGATTGGARTGG-3′) and Reg1320R (5′-AATTGCAAAGATCTATCCC-3′) primers [18]. A total of 50 μL PCR mixture containing 5 μL 10 × Taq Buffer, 1 μL dNTP Mixture (10 mmol/L), 0.25 μL Dream Taq DNA Polymerase (5 U/μL, Mg2 + plus), 1.25 μL of the upstream and downstream primers (10 μmol/L), 1 μL genome DNA (50 ng/μL) was supplemented with ddH2O. The PCR reaction program was as follows: 98 °C for 1 min (1 cycle), 98 °C for 10 s, 50 °C for 30 s, 72 °C for 30 s (35 cycles), and 72 °C for 5 min. The PCR products were assessed using 2% agarose gel electrophoresis, and images of the gel were taken (Gene Amp 9700, Applied Biosystems (ABI), Waltham, MA, USA). The library was constructed using the Ion Plus Fragment Library Kit 48 rxns library (ThermoFisher, Scientific, Waltham, MA, USA). The constructed library was subjected to Qubit quantification and then library testing, and then sequenced using Ion S5TMXL (ThermoFisher, Scientific, Waltham, MA, USA).

### 2.3. Amplicon Gene Sequence Processing

Singletons and low-quality reads were removed from raw reads using Cutadapt (v1.9.1) (http://code.google.com/p/cutadapt/) [19]. According to 97% similarity, clean sequences were clustered into operational taxonomic units (OTUs) using the UPARSE method [20]. Using the Mothur method [21], annotation analysis of OTUs was performed to obtain representative species and count the numbers of OTUs in the taxonomic levels (kingdom, phylum, class, order, family, genus, and species) per sample. Then, the OTUs of bacteria and protozoa were identified and assigned to the bacteria and protozoa SILVA database (version 11.9) (http://www.arb-silva.de) [22]; the OTUs of fungi were assigned in the database of UNITE Release 7.0 database (http://unite.ut.ee) [23]. 

### 2.4. Volatile Fatty Acids

After thawing at 4 °C for 2 h, rumen fluid samples were centrifuged for 10 min at 4 °C with 12,000 rpm. Then, 1.5 mL of supernatant was transferred to 2 mL centrifuge tubes containing 200 μL 25% metaphosphoric acid. After incubating overnight at 4 °C, samples were centrifuged for 15 min at 4 °C at 13,000 rpm. A total of 1 mL supernatant was absorbed into the 1.5 mL centrifuge tube containing 200 μL 0.2458% crotonic acid. After incubation at 4 °C for 4 h, 800 μL of the mixture was transferred into vials for further analysis by gas chromatography. The VFA concentrations were measured using an Agilent Technologies 7890A GC system (Agilent Technologies, Santa Clara, CA, USA) as follows: the injector and FID detector temperatures were set at 200 °C, the column temperature was maintained at 100 °C for 1 min, then the temperature increased to 145 °C at 3 °C/min, followed by a temperature increase from 145 to 200 °C at 20 °C/min, the temperature was maintained at 200 °C for 6 min [24].

### 2.5. Data Analysis

The differences in Alpha diversity and VFA were calculated by One-way ANOVA using SPSS v18. The Unifrac distance was computed by QIIME v1.9.1 (http://qiime.sourceforge.net/) [25]. Bar and box graphs were plotted by Graph Prism v6 (https://www.graphpad.com/). The pie chart was plotted by Microsoft Office Excel 2010 and was based on the relative abundance of microbiota. The network, principle coordinate analysis (PCoA) and LDA effect size (LEfSe) graphs were plotted using the Major bio I-Sanger Cloud Platform (https://www.i-sanger.com/), in which the Linear discriminant analysis (LDA) threshold was higher than 2.5.n

## 3. Results

### 3.1. An Overview of Ruminal Bacteria, Fungi, and Protozoa Composition

The ruminal bacteria contained 2283 OTUs by 16S rRNA gene sequencing. In addition, 24 bacteria phyla were detected, and the prevalence and abundance of each phylum were different among individuals (Figure 1D, Appendix A). Moreover, 15 phyla appeared in all individuals, in which Firmicutes (50.6% in goats, 50.3% in sheep) and Bacteroidetes (41.2% in goats, 40.6% in sheep) were the most abundant phyla. The relative abundance of Actinobacteria ranged from 0.48% (goat_5) to 6.05% (sheep_5). Epsilonbacteraeota was less prevalent and not detected in 11 samples (Appendix A). Of the OTUs, about 90.5% in goats, 88.9% in sheep and 99.8% in goats, 100% in sheep obtained from the core microbiota (Firmicutes and Bacteroidetes), were enriched in Clostridiales and Bacteroidales orders (Appendix A). The major orders for other phyla with the relative abundance ≥1% were Coriobacteriales (Phylum: Actinobacteria, 90.0% in goats, 89.7% in sheep), Spirochaetales (Phylum: Spirochaetes, 100% in goats, 100% in sheep) (Appendix A). At the genus level (the relative abundance ≥1%), the abundance of both Rikenellaceae_RC9_gut and Prevotella_1 belonged to Bacteroidetes (24.5% in goats, 22.6% in sheep) (Figure 1D, Appendix A). By ITS rRNA gene sequencing, 1398 OTUs were obtained, including, Neocallimastigomycota (51.6% in goats, 77.6% in sheep) and Ascomycota (36.0% in goats, 18.0% in sheep) were the core fungi in 13 phyla (Figure 2D, Appendix A). Blastocladiomycota, Cercozoa and Kickxellomycota were detected in one sample. The relative abundance of Neocallimastigomycota was highly different between samples (sheep_5: 89.65% to goats_3: 32.13%) (Appendix A). Neocallimastigaceae (Phylum: Neocallimastigomycota, 100% in goats and sheep), Pleosporales (Phylum: Ascomycota, 60.8% in goats, 55.6% in sheep) and Filobasidiales (Phylum: Basidiomycota, 38.0% in goats, 45.5% in sheep) were the dominant orders (Appendix A). At the genus level, the abundance of unclassified_f__Neocallimastigaceae, Orpinomyces, Caecomyces and Piromyces, which all belonged to Neocallimastigomycota, was the highest (Figure 2D, Appendix A). For protozoa, 270 OTUs were obtained from the rumen of all goats and sheep. Ciliophora, the core microbiota (85.9% in goats, 97.8% in sheep), was unique in 6 phyla, which presented in all individuals (Figure 3D, Appendix A). The OTUs of Ciliophora were enriched in the order Trichostomatia (Appendix A). Ciliophora contained only 14 OTUs. Genus with the relative abundance ≥1% belonged to Ciliophora, in which Metadinium had the highest abundance (58.6% in goats, 75.0% in sheep) (Figure 3D, Appendix A).

### 3.2. Different Composition of Ruminal Bacteria, Fungi, and Protozoa in Goats and Sheep

Goats and sheep shared 1762 OTUs for bacteria. However, 398 and 123 OTUs were not shared in the rumen of goats and sheep, respectively (Figure 1A). For fungi, goats possessed 706 unique OTUs, while sheep possessed 236 unique OTUs, and shared 456 OTUs (Figure 2A). For protozoa, goats contained 224 unique OTUs, while sheep contained 15 unique OTUs, and shared 31 OTUs (Figure 3A). The number of OTUs for bacteria, fungi and protozoa was higher in the rumen of goats. Bacterial alpha diversity was higher in goats than sheep based on the Shannon index (*p* = 0.01) (Figure 1B). However, the alpha diversity of fungi and protozoa was not significantly different between goats and sheep (Figure 2B and Figure 3B). Subsequently, the Bray-Curtis distance was used for whole microbiota abundance. The PCoA plot showed a distinct clustering of bacteria (ANOSIM: R = 0.46; *p* = 0.01) and fungi (ANOSIM: R = 0.37; *p* = 0.04) composition between goats and sheep (Figure 1C and Figure 2C). However, such a pattern was not observed in the PCoA of protozoa (ANOSIM: R = 0.03; *p* = 0.32) (Figure 3C).

Based on LEfSe analysis, comparing the abundance of bacteria between goats and sheep, the phylum Actinobacteria was higher in sheep, and the family F082 was higher in goats (Figure 1E). At the genus level (the relative abundance ≥ 1%) (Appendix A), the relative abundance of norank_f_F082, Saccharofermentans, Lachnospiraceae_XPB1014, and unclassified_f_Rikenellaceae were higher in the goat group, whereas the relative abundance of Olsenella was increased in sheep (Figure 1E). At the OTU level, 164 OTUs of ruminal bacteria were significantly different between the two groups, in which the abundance of 132 OTUs was increased in the goat group, and 32 OTUs were increased in sheep group. The core phylum encompassed 129 OTUs (Bacteroidetes: 51 OTUs; Firmicutes: 78 OTUs) (Appendix A). For fungi, the relative abundance of the phylum Neocallimastigomycota was higher in the sheep group (Figure 2E). Taxa with a high abundance was not significantly different at the family level (Figure 2E, Appendix A). Moreover, at the genus level with high abundance, the relative abundance of Cyllamyces was higher in the sheep group (Figure 2E, Appendix A). A total of 29 OTUs was significantly different, in which only 3 OTUs were increased in the sheep group. In all OTUs with significant differences, OTU145, OTU99, and OTU1108 represented the highest abundance, and all belonged to the phylum Neocallimastigomycota (Appendix A). In the protozoa, OTU22 was only observed in goats and belonged to the phylum Streptophyta (Appendix A). No significant differences were observed at other taxonomy levels.

### 3.3. Rumen Bacteria and Fungi Network between Goats and Sheep

For bacteria, the ruminal bacterial network of goats was constituted of top 25 genera with 24 edges, in which 3 edges represented a negative correlation. According to degree centrality, closeness centrality, and betweenness centrality, Succiniclasticum and Prevotella_1 were the core genus. Three clusters were closely related to Prevotella_1, and Lachnospiraceae_XPB1014 negatively correlated with Prevotella_1. Succiniclasticum and Prevotellaceae_UCG-003 positively correlated with Prevotella_1. Four clusters were closely related to Succiniclasticum, norank_f__F082, Prevotella_1, Ruminococcaceae_UCG-010, and Ruminococcaceae_UCG-014, which positively correlated with Succiniclasticum (Figure 4A). The network of sheep rumen which constituted of 46 edges was more complicated. Only 19 edges represented a negative correlation. Ruminococcaceae_NK4A214 and Butyrivibrio_2 were the core genus. Six clusters closely related to Ruminococcaceae_NK4A214, Saccharofermentans, Prevotellaceae_UCG-001, unclassified_o_Clostridiales, norank_f_F082, and norank_f_Bacteroidales_UCG-001, which negatively correlated with Ruminococcaceae_NK4A214; Butyrivibrio_2 positively correlated with Ruminococcaceae_NK4A214. Eight clusters were closely related to Butyrivibrio_2, norank_f_Muribaculaceae, Ruminococcus_1, Lachnospiraceae_NK3A20, Olsenella, Saccharofermentans and Prevotellaceae_UCG-001, which positively correlated with Butyrivibrio_2; Christensenellaceae_R-7 and norank_f_Bacteroidales_BS11 negatively correlated with Butyrivibrio_2 (Figure 4A). The network of fungi had 45 edges in the rumen of goats and 41 in the rumen of sheep. Moreover, Neocallimastigaceae was the core genus of goats and negatively correlated with other fungi (Dothideomycetes, Pleosporales, Didymellaceae Naganishia, Ascomycota, Sporormiaceae, and Plenodomus) (Figure 4B), Mortierella was the core genus of sheep and positively correlated with other fungi (Gibberella, Plectosphaerella, Guehomyces, Fusarium, Sordariomycetes, Nectriaceae, Alternaria, and Saitozyma) (Figure 4B).

In this study, the results of amplicon sequencing showed that most of the remarkably different bacteria were related to the production of VFA. Therefore, we measured the concentrations of VFA in the rumen to compare the rumen fermentation capacity between goats and sheep. Notably, when compared to sheep, the concentration of acetic acid, propionic acid, and TVFA was significantly increased in the goat rumen (*p* < 0.01). Other VFAs did not show a significant difference (Figure 5).

## 4. Discussion

In this study, rumen microbes were characterized in goats and sheep that lived in the same pasture. Both diversity and composition of rumen microbiota were significantly different in goats and sheep, and the differences in rumen microbe diversity and composition may be driven by the food preferences of goats and sheep.

### 4.1. Feature of Rumen Microbiota in the High-Altitude Pasture

In all goats and sheep (n = 5 each), a diversity of 24 bacteria phyla (2283 OTUs), 13 fungi phyla (1398 OTUs), and 6 protozoa phyla (270 OTUs) was detected. In addition, microbial diversity was highly observed in high-altitude ruminants [26,27]. The rarefaction curves of bacteria, fungi, and protozoa (Appendix A) suggested that the number of OTUs was sufficiently captured and that the sequencing depth of the samples was sufficient. When compared to other studies, this study firstly expounded the prevalence and abundance of bacteria, fungi, and protozoa in the rumen of high-altitude ruminants at various taxonomic levels. One of the main functions of the rumen was cellulose degradation. Especially, Fibrobacter (a fiber-degrading bacteria) was not observed in this study. The Neocallimastigomycota of fungi (producing high fiber degrading enzyme) [28], and the genus Metadinium of protozoa [29] (Figure 2D and Figure 3D, Appendix A) appeared in the rumen with high abundance. Fungi and protozoa might play an important role in the capability of degrading cellulose in the rumen.

The Tibetan Plateau is known for its harsh environment including cold and hypoxic conditions, as well as a low biomass. In order to adapt to the harsh environment, ruminants that live at a high-altitude pasture had evolved low-methane and high-VFA phenotypes [10]. Methane emission is an important energy-consuming pathway, resulting in energy loss of the diet [30]. The increase in VFA could greatly inhibit the production of methane by competing for hydrogen in the methane-producing pathway [31]. In this study, Treponema and Neocallimastigomycota (Figure 1D and Figure 2D, Appendix A), having a H2-production capacity, were present in the rumen at high abundance [32]. Markedly, Prevotella_1, is involved in the metabolization of proteins, peptides, starch, hemicellulose, and pectin, and VFA production (Appendix A and Figure 1D) [33]. Treponema produces acetate from hydrogen as a substrate that appeared in the rumen of goats and sheep with high abundance [34]. Therefore, in the high-altitude rumen, more hydrogen might be involved in the synthesis of VFA than methane emission. Previous studies have indicated that rumen microbiota have played a key role in the adaption of the ruminant to high-altitude pasture. 

### 4.2. Rumen Microbiota May Reflect Host’s Adaptive Capacity of Environment

Alpha diversity was analyzed using One-way ANOVA. The Shannon index was high in goats. PCoA analysis showed that 10 individuals clustered in accordance with the host. A few studies on livestock had shown similar host-related stratification in microbiota. The microbiota in dairy cattle and yellow cattle was significantly different from that in two yak herds that lived at different altitudes [35]. In a study on Yak and Tibetan sheep, it was shown that the prokaryotic community structure between yak and Tibetan sheep was significantly different (*p* < 0.01) [36]. In another study on yak rumen, a high bacterial diversity was observed when compared to Tibetan sheep [37]. These studies considered the host as the major influencing factor but did not consider the differences in feeding habits. In this study, Cyllamyces (the family Neocallimastigaceae) was related to fiber degradation [38], which was increased in the rumen of sheep (Figure 2E). Saccharofermentans fermented several carbohydrates, and mainly produced propionic acid, which was increased in the rumen of goats (Figure 1E) [39]. Moreover, the concentration of propionic acid was higher in goat rumen (Figure 5). The high abundance of Saccharofermentans indicated the high level of crude protein in the diet [40]. Also, a high crude protein diet significantly increased the growth performance of ruminants under grazing conditions [41]. Lachnospiraceae_XPB1014 produced an array of bacteriocins and butyrate [42], which was increased in the rumen of goats (Figure 1E). In this study, no significant differences were observed in the concentration of butyrate acid between goats and sheep (Figure 5). The reason might be that Lachnospiraceae_XPB1014 was not just involved in butyrate production. The increase of fiber-degrading microbiota caused more hydrogen production in the rumen of sheep compared to goats. However, the decrease in VFA-producing microbiota indicated higher methane emissions and more energy consumption in the rumen of sheep. In a previous study, it was suggested that methanogens attach to protozoa and fungi (which degrade fibers to produce hydrogen) to obtain hydrogen [43]. 

VFA played an important role in ruminant growth and immunity. VFA could make intestinal epithelial cells highly keratinized, which provided a physical barrier for the rumen environment [44]. Acetic and propionate acid could activate the GPR43 and GPR41 receptor to product PYY and GLP-1 hormones which increased glucose utilization for body energy [45]. Notably, acetic acid, the major VFA in the rumen, could decrease the abundance of Escherichia coli to maintain rumen health [46]. In this study, TVFA, acetic acid, and propionate acid were significantly increased in goat rumen, suggesting that goats were more suitable to the harsh conditions of Tibetan pastures compared to sheep.

### 4.3. The Role of the Diet in Rumen Microbiota of Goats and Sheep

In this study, the diversity and composition of rumen microbiota were different between goats and sheep that lived in the same pasture. A study on the gut microbiome of pigs during nursing and weaning suggested that the diet played a key role in shaping the gut microbiome, and that a microbiome with different functions reflected the dietary composition [47]. Similarly, in another study, it was found that the gut microbiome was highly dynamic, exhibiting daily cyclical fluctuations in composition influenced by diet, and it was suggested that diet was the dominant influencing factor for gut microbiota [48]. In our study, we could not eliminate the impact of the host gene, although goats and sheep had the same father by artificial insemination and the same age and sex by estrus synchronization. In a study on primates, it was a discovered that Varecia variegata and Lemur catta had a diverse gut morphology. After being fed a diet of a rotating fruit and vegetable mix, V. variegata and L. catta had similar microbial structures [49]. Several studies on the effect of altitude on rumen microbiota showed that Christensenellaceae R7, Succiniclasticum and Ruminococcaceae highly corrected with altitude [35,50]. These bacteria could secrete various enzymes and be associated with food efficiency [51,52,53]. Notably, the numbers of these bacteria in the rumen depended on the diet [54,55]. The bacteria related to altitude appeared in this study with high abundance (Figure 1D). Interestingly, Alloprevotella, that appeared in the rumen in this study (Appendix A), decreased the risk of cardiovascular disease [56], and the absence of oxygen could lead to cardiovascular disease [57]. Therefore, the role of diets might be more important for the colonization of rumen microbiota. In the case that goats and sheep grazed with a high-intensity in mixed herds and goats reduced dry matter intake and increased feed intake of crude protein. However, the intake of food containing mature fiber forage increased in sheep [58]. We inferred that the difference in rumen microbial diversity and composition between goats and sheep might be driven by feeding preferences. 

## 5. Conclusions

In our study, we characterized ruminal bacteria, fungi, and protozoa at multiple taxonomic levels in the highland pasture, and found differences in rumen microbial diversity and composition between goats and sheep that lived in the same pasture. The family Neocallimastigaceae and the genus Metadinium were enriched in sheep, whereas Saccharofermentans and Lachnospiraceae_XPB1014 were enriched in goats. These differences caused the production of different metabolites between goats and sheep. Furthermore, we found that compared to sheep, the rumen of goats had a higher VFA level. The above-mentioned results suggested that rumen microbiota facilitated goats to better adapt to high-altitude pastures. Therefore, in future studies, we aim to investigate the differences in feeding preference for types of plants between goats and sheep by measuring the concentration of alkanes. Ultimately, we hope to improve ruminant tolerance for the plateau area by diet.

## Figures and Tables

**Figure 1 animals-10-00186-f001:**
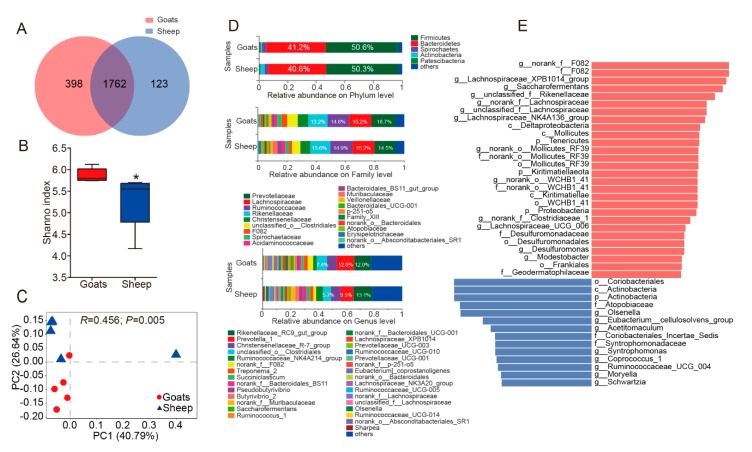
Diversity and composition of ruminal bacteria in goats and sheep. (**A**). Mutual and particular operational taxonomic unit (OTU) number of bacteria in goats and sheep. (**B**). Comparison of the diversity estimation of the bacteria based on One-way ANOVA, in which the *y*-axis represented Shannon index. The Shannon index was related to the richness and evenness of the microbial community. (**C**). Principal coordinate analysis (PCoA) profile of bacteria diversity in all samples using an unweighted UniFrac metric. The percentage of variation explained by PC1 and PC2 were indicated in the axis X and Y. Statistic (R) and *p*-value (P) were annotated by ANOSIM analysis between goats and sheep. (**D**). The illustration exhibited the relative distribution of the most dominant bacteria (1% of the total sequences) in goats and sheep at the phylum, family, and genus level. (**E**). Taxa with significantly different abundances at phylum (p), class (c), order (o), family (f), genus (g) levels are shown. The red color represented that the abundance of taxa was higher in goats and a blue color represented that the abundance of taxa was higher in goats. The higher the Linear Discriminant Analysis (LDA) score, the greater the effect of the taxa abundance on the difference between goats and sheep.

**Figure 2 animals-10-00186-f002:**
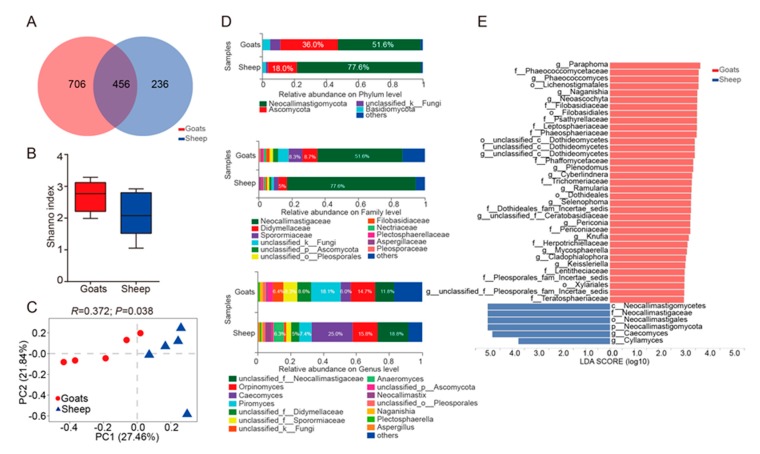
Diversity and composition of ruminal fungi in goats and sheep. (**A**). Mutual and particular operational taxonomic unit (OTU) number of fungi in goats and sheep. (**B**). Comparison of the diversity estimation of the fungi based on One-way ANOVA, in which the *y*-axis represented Shannon index. The Shannon index was related to the richness and evenness of the microbial community. (**C**). Principal coordinate analysis (PCoA) profile of fungi diversity in all samples using an unweighted UniFrac metric. The percentage of variation explained by PC1 and PC2 were indicated in the axis X and Y. Statistic (R) and *p*-value (P) were annotated by ANOSIM analysis between goats and sheep. (**D**). The illustration exhibited the relative distribution of the most dominant fungi (1% of the total sequences) in goats and sheep at the phylum, family, and genus level. (**E**). Taxa with significantly different abundances at phylum (p), class (c), order (o), family (f), genus (g) levels are shown. The red color represented that the abundance of taxa was higher in goats and a blue color represented that the abundance of taxa was higher in goats. The higher the Linear Discriminant Analysis (LDA) score, the greater the effect of the taxa abundance on the difference between goats and sheep.

**Figure 3 animals-10-00186-f003:**
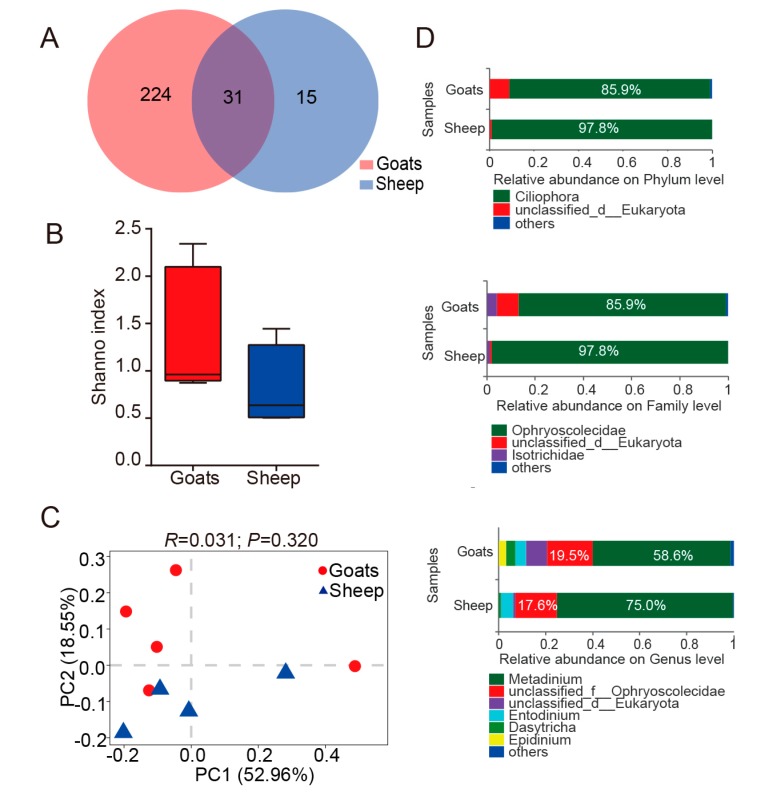
Diversity and composition of ruminal fungi in goats and sheep. (**A**). Mutual and particular operational taxonomic unit (OTU) number of protozoa in goats and sheep. (**B**). Comparison of the diversity estimation of the protozoa based on One-way ANOVA, in which the *y*-axis represented Shannon index. The Shannon index was related to the richness and evenness of the microbial community. (**C**). Principal coordinate analysis (PCoA) profile of protozoa diversity in all samples using an unweighted UniFrac metric. The percentage of variation explained by PC1 and PC2 were indicated in the axis X and Y. Statistic (R) and P-value (P) were annotated by ANOSIM analysis between goats and sheep. (**D**). The illustration exhibited the relative distribution of the most dominant protozoa (1% of the total sequences) in goats and sheep at the phylum, family, and genus level. 3.4 The concentrations of ruminal volatile fatty acids in goats and sheep.

**Figure 4 animals-10-00186-f004:**
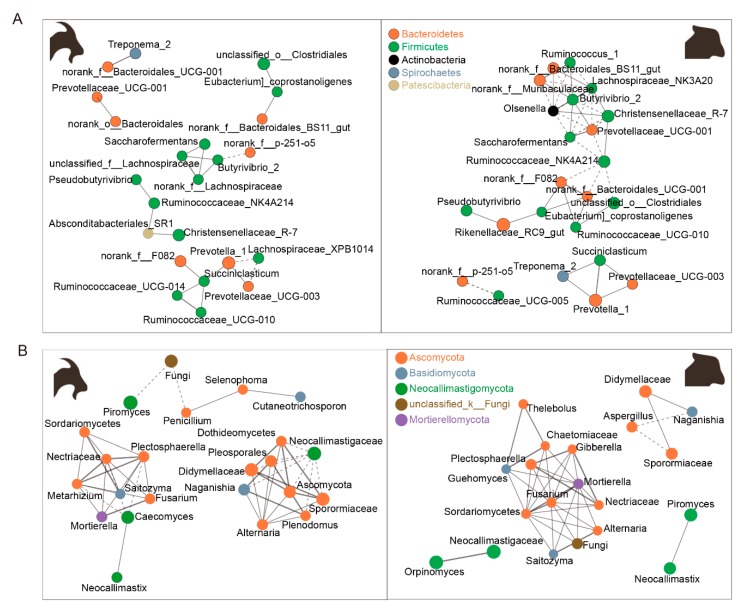
Network analysis of bacteria and fungi. Network analysis applied to microbiota in the rumen of goats and sheep. Genus correlation network maps mainly reflected the genus (relative abundance top 25)-relatedness under the same host. The correlation coefficient between genera reflected the correlation between genera. The size of the node was proportional to the abundance of the genera. Node color corresponded to phylum taxonomic classification. A solid or dotted line of the edge represented positive or negative correlations, respectively and the edge thickness was equivalent to the correlation value. (**A**) bacteria and (**B**) fungi.

**Figure 5 animals-10-00186-f005:**
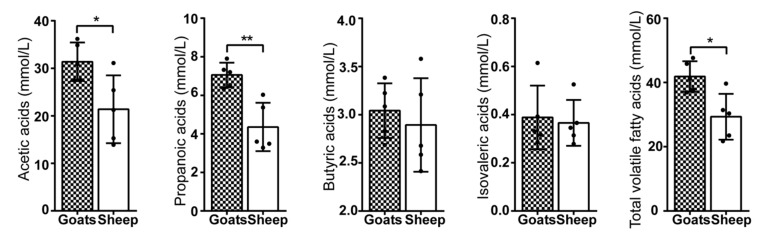
The concentrations of volatile fatty acids. The volatile fatty acids level in rumen and One-way ANOVA analysis between goats and sheep group are shown. * *p* < 0.05, ** *p* < 0.01, *** *p* < 0.001.

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
