# Peer review of "Diversity and Composition of Rumen Bacteria, Fungi, and Protozoa in Goats and Sheep Living in the Same High-Altitude Pasture"

_animals, 2020, doi:10.3390/ani10020186_

Round 1

Reviewer 1 Report

Review of manuscript number: Animals-678012

Title: Diversity and composition of rumen bacteria, fungi and protozoa in goats and sheep living in the same high-altitude pasture

The aim of the study was to comparatively characterize rumen microbiota in Tibetan sheep and goats living at high altitudes.

The study was conducted on 40 young (one-year-old) animals (20 goats and 20 sheep). The animals were grazed in Qiangtang Plateau (Nagchu, Tibet, China; altitude more than 4,800 m). The research material was rumen fluid obtained through rumen fistula.

In order to investigate the diversity and composition of bacteria, fungi, and protozoa in the rumen, amplicon gene sequencing was performed from rumen fluid samples of these both Tibetan species of animals, living at the same pasture in highland.

Differentiation of microbiota species was carried out by DNA extraction and their identification using PCR method. Taxonomic classification was based on operational taxonomic units (OTUs) for bacteria, fungi and protozoa in accordance with the given literature data.

The research required a lot of commitment from the authors; they were well planned and properly conducted. The results are also interesting.

The researches are very interesting because it concerns animals living far away from civilization, according to nature for a long time. These animals can be a reference standard for animals bred on modern farms.

In my opinion, in these studies missing comparison of the values obtained from Tibetan sheep and goats to sheep and goats that are bred in more modern conditions. Even in relation to the literature.

It is very interesting that animals at high altitudes develop microorganisms that produce small amounts of methane and more hydrogen. At the present time, efforts to reduce greenhouse gas emissions in every area of production also in animal husbandry, is very important and such research are needed. Perhaps it is possible to affect (influence) on the microbiota of the digestive tract of animals, towards reducing methane and carbon dioxide production on large industrial farms.

The manuscript is generally well written, I do not notice serious mistakes. However, there are some minor errors or inaccuracies.

Introduction

This part is well written, introduces the reader to the subject and further parts of the work.

The purpose of the research should be clearly defined. Currently, the purpose of the work is unclear, mainly the second part of the sentence L76 “….and discuss the influencing factors of rumen microbiota between different hosts.” maybe better “…and demonstrating of the influencing factors on the quality of the rumen microbiota between different hosts.” Whether this will be changed, depends on the authors.

L 75 Should be… The aim of the study…. not “the”

Material and methods

The animals used for the research, material for tests and the research methods were correctly described.

Results

The form of presentation of the results is unclear; it is difficult to associate the description of the text with tables and figures. Perhaps this is due to the large number of presented results.

I know that it is difficult to clearly show a large number of results (with statistics), but please try to show more clearly, this will increase the value of the manuscript.

Of course, the results presented in current form are also acceptable.

Discussion

L261 Why was the differences in the prokaryotic community structure between Yak and Tibetan sheep? It needs to be clarified (one sentence), maybe it was due to differences in nutrition, maybe they chose different plants.

In my opinion, in the discussion (part 4.3.) lacks comparing the results of own research (Tibetan sheep and goats) with animals (sheep and goats) in lowland farms, this would confirm that different nutrition affects the rumen microbiota. On this subject, there is only one sentence (throughout the manuscript) in the introduction, L 65-67.

Conclusions

In this part, specific conclusions should be provided, which results from own research.

L295-297 this sentence is not true; no such research was carried out in this manuscript it was only discussed based on literature data.

L297-299 this sentence is not clear. These findings may help to explaining how animals adapt to high-altitude grazing conditions.

Reviewer 2 Report

To my opinion, the paper provides some interesting information. However, I have some serious issues with study design, description of methods, and discussion of the results.

Overall: please provide English editing by a native speaker or professional service; there are a lot of spelling and wording problems. It may be valuable to solely describe microbial communities in a certain environment, but this makes the results very specific. In the case of this study, you compared sheep and goats, which is okay, but for me, a valuable study design would have included a group of sheep and goats grazing on low-altitude pasture in order to assess the effect of altitude. Moreover, analyses of microbial metabolites would have increased validity of your study. Simple Summary: please be more specific; you wrote ". . . increase . . ." here and at several other points in the text: what do you mean, what have increased compared to what? Line 52-53: a reference is missing. Introduction: please revise: the introduction only belongs to goats, information on sheep are missing. There are some statements that need further explanation: line 56-59, line 62-65, line 65 (explain why), and line 67-71 (not clear what you want to say; explain the link between tannins and your study objectives). Materials and Methods: a convincing ethical statement is missing! Please clearly describe the process of cannulation of the animals. Where the test animals of both sexes? Did you use technical replicates of ruminal fluid samples? Line 116-118: the sentence ends incomplete. Section 4.3: nice examples, but I would rather stay in the ruminant sector. Conclusions: again, be more precise, sum up the key findings, and clearly state the implications of your results. Line 296-297: where did you present these data?

Round 2

Reviewer 2 Report

A convincing revision was provided. Just a few minor issues:

L129: To . . . seems incorrect.

L146-147: what means "post ran for 1 min"?

L207: THE?

Fig. 2: is missing at this point.

L233 and following lines: this section is formatted incorrectly.

L256: is this Fig. 3? a description is missing.

L257: fatty instead of fattey.

L261: propanoic or propionic acid?

L299: space is missing between rumen and microbiota.

Fig. 5: description should appear directly under the figure; y-axis: should read acetic . . . acid not acids.

I could not open the supplementary files, thus could not review it.
